# Effect of Alloying Elements and Low Temperature Plasma Nitriding on Corrosion Resistance of Stainless Steel

**DOI:** 10.3390/ma15196575

**Published:** 2022-09-22

**Authors:** Yanjie Liu, Daoxin Liu, Xiaohua Zhang, Wenfeng Li, Amin Ma, Kaifa Fan, Wanzi Xing

**Affiliations:** 1College of Civil Aviation, Northwestern Polytechnical University, Xi’an 710072, China; 2No. 5702 Factory of Chinese People’s Liberation Army, Xianyang 712201, China

**Keywords:** martensitic stainless steel, low temperature plasma nitriding, density functional theory, corrosion, electrochemical characteristics

## Abstract

Although nitriding treatment usually improves the hardness and wear resistance of stainless steel, it also reduces its corrosion resistance. The effects of different nitriding temperatures and time and main alloying elements in stainless steel on the properties of the martensitic precipitation hardening of stainless steel were studied by first-principles calculations and experiments in this study. The results showed that the corrosion resistance of the martensitic stainless steel 0Cr17Ni4Cu4Nb was much lower than that of 1Cr15Ni2Mo2Cu before and after nitriding. According to the density functional theory calculation results, the molybdenum-containing stainless steel had higher stability and corrosion resistance and a lower Fermi level, electron conduction concentration and electrochemical activity than the niobium-containing stainless steel before and after nitriding. In addition, at the same temperature, the surface hardness of the 1Cr15Ni2Mo2Cu steel increased linearly with the prolongation of nitriding time, but its corrosion resistance decreased. Under the same nitriding time (24 h), the nitriding temperature increased from 300 to 450 °C, and the surface hardness and nitriding layer depth of the nitriding steel increased gradually, while the corrosion resistance decreased gradually. These results were attributed to the Cr-poor phenomenon caused by the formation of CrN. The 1Cr15Ni2Mo2Cu martensitic stainless steel obtained a high surface hardness after nitriding at 300 °C for 24 h, and the corrosion resistance did not decrease.

## 1. Introduction

Martensitic stainless steel has good impact and corrosion resistance and high plasticity [1,2], and is usually used to manufacture steam turbine blades, valves, oil pipelines, knives and medical devices [3,4]. However, martensitic stainless steel has low hardness and poor wear resistance, which limits its industrial application. Chemical heat treatment of the surface of stainless steel can effectively improve its surface hardness and wear resistance [5]. Nitriding treatment is one of the most commonly used methods to improve the wear resistance of materials. The nitriding treatment of austenitic stainless steel at 500 °C can significantly improve the surface hardness and wear resistance on the thicker nitride layer on the surface [6].

Dong et al. [7] showed that 17-4PH stainless steel is ion nitrided at a low temperature of 420 °C, and an expanded martensitic phase is formed in the nitrided layer, with its hardness and wear resistance significantly improved. Li et al. [8] modified ion nitriding on 17-4PH stainless steel. By studying the microstructural characteristics of the nitriding layer at different temperatures, it was found that the tribological properties of 17-4PH stainless steel were improved after the nitriding treatment. However, after the traditional plasma nitriding process, CrN and Cr_2_N precipitate out of the stainless steel, which improves the surface hardness of the material and seriously reduces the corrosion resistance [1,9]. Nevertheless, the solid solution of a small amount of elemental N can avoid the formation of iron oxide and help to improve the pitting corrosion resistance of stainless steel [10]. However, excessive N will produce a Cr-rich nitride zone in stainless steel, and the Cr content in the stainless steel will be significantly reduced, preventing the stainless steel from forming a passive film on the surface, which destroys its corrosion resistance. To solve this problem, Zhang and Bell invented the low-temperature plasma nitriding process, which not only improves the wear resistance but also maintains the corrosion resistance of stainless steel [11]. It is found that austenitic stainless steel can produce supersaturated nitrogen-containing austenite after low-temperature plasma nitriding, causing lattice distortion of austenite and producing the wear- and corrosion-resistant S phase [12,13,14]. The elemental N of the S phase is dissolved in the austenite, which inhibits the precipitation of CrN at the grain boundary and improves the wear resistance to ensure corrosion resistance.

However, martensitic stainless steel makes it easy to form CrC during heat treatment and CrN during nitriding, leading to further consumption of corrosion-resistant elemental Cr [15,16,17,18,19]. It is found that the Cr content in steel must be greater than 10.5 wt.% to ensure the corrosion resistance of steel [20]. Therefore, with the consumption of Cr, the corrosion resistance of martensitic stainless steel will inevitably decline. Current research shows that the formation of CrN can be reduced or avoided by regulating the nitriding process to maintain the corrosion resistance of stainless steel. Some researchers found that AISI 420 stainless steel was nitrided for 5 h at 420 °C, and CrN was not initiated in the nitride layer. The test results showed that the wear resistance of stainless steel was increased by three or four times after nitriding, and the corrosion resistance did not significantly decrease [21]. Sun et al. [22] found that for 17-4PH stainless steel, 425 °C is the transition temperature of the image composition of its nitrided layer. When the nitriding temperature is lower than 425 °C, the X-ray diffraction (XRD) pattern of the nitride layer of 17-4PH stainless steel is amorphous. When the nitriding temperature is higher than 425 °C, the XRD pattern of the nitride layer is composed of CrN and γ-Fe_4_N. The corrosion resistance of Fe_4_N decreases significantly due to its mixed phase composition. Furthermore, it has been found that ε-Fe_3_N and γ′-Fe_4_N, and α‘_N_ (nitrogen-expanded martensite) that is formed after martensitic stainless steel nitriding, are beneficial to the corrosion resistance of materials [19].

In conclusion, the consumption of Cr in stainless steel could be avoided through a reasonable nitriding process to maintain corrosion resistance. However, most of the above studies were based on the adjustment of the nitriding process, ignoring the influence of stainless steel composition on the nitriding process and corrosion resistance after nitriding. Therefore, in this study, we compared the new stainless steel 1Cr15Ni2Mo2Cu with the traditional stainless steel 0Cr17Ni4Cu4Nb. The reason for choosing these two kinds of stainless steel is that 1Cr15Ni2Mo2Cu is a new stainless steel product, which uses the element Mo to replace the element Nb in 0Cr17Ni4Cu4Nb, and slightly adjusts the content of other alloy elements, leading to different mechanical properties, but they are candidate materials for steam turbines. In addition, Mo has strong pitting corrosion resistance and Nb has strong grain boundary corrosion resistance in stainless steel. These two elements may affect the corrosion resistance of materials. Combined with density functional theory (DFT) calculations and experiments, the effects of the stainless steel constituent elements and different low temperature nitriding processes on the corrosion resistance of the two kinds of stainless steel were studied.

## 2. Experimental and Numerical Simulation Methods

### 2.1. Experimental Methods

#### 2.1.1. Experimental Materials and Sample Preparation

The 1Cr15Ni2Mo2Cu (1Cr15) and 0Cr17Ni4Cu4Nb (17-4PH) martensitic precipitation hardening stainless steels were used and their chemical compositions are shown in Table 1. The difference between these two kinds of stainless steels was whether they contained molybdenum or niobium. Before processing, both kinds of stainless steel were treated with solid solution and double aging treatments. The mechanical properties of the stainless steels are shown in Table 2.

The 1Cr15 and 17-4PH stainless steel rods were processed into ϕ 24 × 8 mm disc specimens. Before nitriding, the surface was polished with 1000# SiC sandpaper on both sides of the sample, and mechanical polishing was conducted. Finally, the samples were ultrasonically cleaned with a metal decontamination powder and ethanol solution.

#### 2.1.2. Low Temperature Plasma Nitriding Treatment

The specimen was placed on the sample table shown in Figure 1 and connected to the cathode of a pulsed power supply. A vacuum was pumped to below 5 Pa, and Ar gas was passed through the sample for glow sputter cleaning for 1 h. The atmosphere was then replaced with 80% N_2_ + 20% Ar, and the pulsed power supply was switched on, keeping the voltage at −600 V, the duty cycle in the range of 35–55% and the frequency fixed at 10 kHz. Temperatures of 250, 300, 350, 400 and 450 °C were selected, respectively, for 1Cr15 stainless steel, which was subjected to plasma nitriding treatment for 24 h. In addition, plasma nitriding treatment at 300 °C for 48 h provided a simultaneous study of the effect of nitriding time on the material properties. In contrast, 17-4PH was subjected to plasma nitriding at 300 °C for 24 and 48 h to explore the effect of different alloy elements on the corrosion resistance of the materials.

#### 2.1.3. Characterization

In this study, scanning electron microscopy (SEM, JSM-6390A) and an X’pert-PRO X-ray analyzer were used to analyze and observe the surface morphology and the structure of each phase before and after the experiment. Phase identification used Cu-Kα radiation (λ = 1.5406 Å) as a radiation source, a tube voltage and current of 35 kV and 40 mA, scanning between 20 and 90 deg (2θ), at a scanning rate 10 °/min.

An HV-1000 microhardness tester with a Knoop indenter was used to test the surface hardness change in the sample with a 0.245 N load maintained for 20 s. During measurement of the hardness, five points were taken along the horizontal direction, then the average value was taken as the hardness of the surface or a certain depth.

Electrochemical impedance spectroscopy (EIS) and kinetic potential polarization curves were measured using a PARSTAT2273 electrochemical comprehensive testing system. A three-electrode system was adopted, with the saturated calomel electrode as the reference electrode, the platinum electrode as the auxiliary electrode, and the sample as the working electrode. The effective exposure area was 1 cm^2^. The test temperature was 35 ± 2 °C, and the electrolyte was a 3.5% NaCl solution. After connecting the equipment, the open circuit potential time curve was measured. When the corrosion potential was stable, the AC impedance spectrum in the frequency range of 1000 kHz–10 mHz was measured with an excitation voltage of ±10 mV. The polarization curve was determined at a scanning speed of 1 mV/s.

An SY/Q-750 salt spray test chamber was used for the corrosion resistance test. The corrosion medium was a 5% NaCl solution, the pH value was maintained between 6.5 and 7.2, the relative humidity was maintained at 96 ± 2%, and the temperature was 35 ± 2 °C. The corrosion resistance was evaluated by observing the surface morphology characteristics of the samples.

### 2.2. First-Principles Calculations

The Vienna Ab-initio Simulation Package (VASP) [23,24,25,26] software based on DFT was used for the calculations. The exchange–correlation function used in the calculations was processed by Perdew–Burke–Ernzerhof (PBE) [27] under the generalized gradient approximation [28]. The electron–particle interactions were described using the projected augmented plane-wave method [29]. The cut-off energy was chosen to be 450 eV, the Brillouin zone K-point grid was set to 2 × 2 × 2 by the Monkhorst–Pack method [30], the energy convergence accuracy of the electron step was 1.0 × 10^−4^ eV/atom, and the overall atomic force convergence threshold of the geometry optimization process was 2.0 × 10^−2^ eV/Å.

The martensitic stainless steel has α-Fe as the main lattice structure, and the crystal belongs to the cubic IM-3M space group, with two Fe atoms per cell. Fe, Cr, Nb and Mo atoms tend to form substitutional solid solutions due to their similar atomic radii. Therefore, we studied the effect of elements such as Mo or Nb on the properties of martensitic stainless steel. The actual content of Cr in 1Cr15 and 17-4PH stainless steel was ~16 wt.%. Therefore, we constructed 3 × 3 × 3 α-Fe supercells to simulate the solid solutions.

To ensure that the simulated Cr content was consistent, we used 22 Cr atoms and another metal atom instead of Fe atoms to construct solid solutions of Fe105Cr22Mo and Fe105Cr22Nb for the calculations (Figure 2). The Cr content of the solid solution formed was ~16 wt.%, which is close to the actual content in stainless steel. This could be used to effectively study the effect of Mo and Nb on the properties of martensitic stainless steel. In addition, to fill the octahedral gap of the above solid solution model with different numbers of N atoms, we formed Fe105Cr22MNx (x = 0, 1, 2 or 4; M = Mo or Nb) solid solution and explored the effect of nitriding on stainless steel.

In order to measure the stability of the alloy, its cohesive and formation energies were calculated. Alloy cohesive energy refers to the energy released from alloy formation from isolated atoms. A lower cohesive energy means that the system is more stable.

The cohesive energy equation is as follows:$$E_{\mathrm{coh}}=\frac{1}{N_{tot}}(E_{tot}-\sum_{i}^{k}n_{i}E_{atom})$$
where $N_{tot}$ represents the total number of atoms in the solid solution, $E_{tot}$ is the total energy of the solid solution, i represents the i atom in the solid solution, k represents the type of total elements in the solid solution, $n_{i}$ represents the number of atoms of the i element in the solid solution, and $E_{atom}$ is the energy of the isolated electrons of i element.

The formation energy of the alloy refers to the energy absorbed and released by metal atoms from simple substances to form compounds. A negative formation energy indicates that the alloy can easily form. Otherwise, it needs to absorb energy to form. The formation energy equation is as follows:$$E_{\mathrm{form}}=\frac{1}{N_{tot}}(E_{tot}-\sum_{i}^{k}n_{i}E_{norm})$$
where $N_{tot}$ represents the total number of atoms in the solid solution,$\;E_{tot}$ is the total energy of the solid solution, i represents the i atom in the solid solution, k represents the type of total elements in the solid solution, $n_{i}$ represents the number of atoms of the i element in the solid solution, and $E_{norm}$ is the normalization energy of the i element with a stable simple substance.

## 3. Results and Discussion

### 3.1. DFT Results

The total, formation and cohesive energies of different replacement solid solutions are shown in Table 3. The formation energy of stainless steel after alloying was negative. Table 3 shows that both elements can exist stably in stainless steel. Moreover, comparing the cohesion energy, it can be seen that the stainless steel with Mo was more stable.

As shown in Figure 3, the formation and cohesion energies were negative for all stainless steels after nitriding. It can be seen from the formation energy curve that the formation energy of stainless steel after Mo or Nb alloying decreased with increasing N content (Figure 3a). The N solubility of martensitic stainless steel could be improved to a certain extent after alloying. From the cohesion energy curve, when a small amount of N atoms were dissolved in the solid solution, the steel containing Nb had higher stability than that containing Mo stainless steel. However, with the gradual increase in N content, the stainless steel containing Mo was more stable (Figure 3b). These results showed that the low N solubility of martensitic stainless steel could be improved by adding these two elements. The stainless steel containing Mo had higher stability than the stainless steel containing Nb before nitriding and after nitriding for a long time.

From an electrochemistry perspective, metal corrosion is related to electron transfer. The electron transfer process is the essence of the redox reaction in the corrosion process. The electrochemical corrosion process must have an electronic and ionic conductive path. According to the existing theory, the Fermi level is a metal’s critical pitting potential energy [31]. The Fermi energy level of materials is closely related to their electrode potential. The higher the Fermi level of a system, the easier it is to lose the outer electrons, and the more prone it is to corrosion. Therefore, by judging the stability of the electronic structure and the Fermi energy level, the difference in the corrosion resistance of the alloy could be analyzed [32].

It can be seen from Table 4 that for structures without N, the Fermi level of Fe105Cr22Mo was lower than that of Fe105Cr22Nb. For the structure containing N, the Fermi level of any alloying stainless steel increased gradually with the addition of N atoms, and the material became increasingly less corrosion resistant. Table 4 also reflects the changing trend consistent with Figure 3, i.e., at low nitrogen concentrations, the corrosion resistance of stainless steel containing Nb was better than that of stainless steel containing Mo, but with increasing N content, the Fermi level of stainless steel containing Mo began to trend lower than that of stainless steel containing Nb. Table 4 indicates that adding Mo could give the stainless steel substrate and stainless steel after nitriding better corrosion resistance.

The density of states (DOS) reflects the distribution of electrons in a certain energy range. It can be seen from the DOS diagram calculated in Figure 4 that the density of states of electrons near the Fermi level of Mo alloyed stainless steel was lower than that of Nb alloyed stainless steel under any nitriding degree. The DOS of electrons at the Fermi level represents the concentration of conduction electrons in the corrosion process [33]. Therefore, compared with the Nb element, the addition of the Mo element reduced the electron conduction concentration in the material’s corrosion process and reduced the material’s electrochemical activity as a whole. This is also consistent with the research results of Sun, which showed that the passivation current density of stainless steel containing Mo was significantly lower than that of molybdenum-free stainless steel [34]. Meanwhile, the calculated results were also the same as the experimental electrochemical measurement results. The self-corrosion potential of 1Cr15 stainless steel containing Mo is lower than that of 17-4PH stainless steel containing Nb. In conclusion, Mo alloying could significantly improve the structural stability while maintaining a low Fermi level, reducing the overall electrochemical activity of stainless steel and enhancing the solubility of N in steel. Therefore, the stainless steel containing Mo had better corrosion resistance, and the corrosion resistance was still stronger than that of the stainless steel containing Nb after the nitriding treatment.

### 3.2. Experimental Results

#### 3.2.1. Surface Morphology and Microstructure

Figure 5 and Figure 6 show that the surfaces of the samples of 1Cr15 stainless steel nitrided below 350 °C were relatively flat (Figure 5a,b). However, large particles appeared on the 17-4PH samples nitrided at 300 °C for 48 h and 1Cr15 samples nitrided at 350 °C for 24 h, as shown in Figure 5c and Figure 6d. According to the analysis of Figure 7c, it generated CrN on the surfaces of ε-Fe_3_N and γ-Fe_4_N phases. As shown in Figure 5c, nitride particles were preferentially formed at grain boundaries and distributed with grain boundaries. When the nitriding temperature increased, the diameters of spherical nitride particles on the sample’s surface became larger, the number increased, and nitrides appeared as agglomerations. Through EDS analysis of Figure 5c, it can be seen that the nitrogen content of the crystal on the surface of the sample was significantly higher than that of the flat part (Table 5). This indicated that the nitrogen concentration required for nitride precipitation on the surface of 1Cr15 stainless steel was between 6.22 and 9.67 wt.%. At the maximum nitriding temperature of 450 °C, the nitride crystal particles on the surface of the sample showed cauliflower-like characteristics. Simultaneously, with the extension of nitriding time, there were increasingly more nitride particles on the surface. It could be inferred from the surface morphology that with the increase in nitriding temperature and nitriding time, the sample’s surface became increasingly rougher.

The phase composition of the stainless steel base metal (BM) and nitride layer was analyzed by XRD. The XRD curves had no amorphous dispersion peaks, so the BM and nitrided sample were in the crystalline state. Figure 7a shows that the BM of 1Cr15 and 17-4PH stainless steel was mainly martensitic with a certain amount of γ-Fe phase. Figure 7b shows that 1Cr15 α-Fe was continuously consumed to generate ε-Fe_3_N and γ-Fe_4_N until the 450 °C α-Fe phase disappeared completely.

It is well-known that Cr in stainless steel can effectively improve the corrosion resistance because it can combine with oxygen in the corrosive environment, forming a dense and continuous passive film on the surface of the material. The N element can stabilize austenite and enhance the ability of stainless steel to resist pitting and crevice corrosion [35,36]. However, once these two elements are combined, the CrN phase will appear, which will lead to large consumption of the corrosion-resistant Cr in the stainless steel solid solution and difficulty in forming a passive film on the surface, resulting in the Cr-poor phenomenon of stainless steel and affecting the corrosion resistance of the material. The analysis showed that CrN phase precipitation occurred in the 1Cr15 stainless steel nitriding sample at 300 °C for 48h, and the CrN content increased with nitriding temperature. Therefore, it can be speculated that the corrosion resistance of the sample will be reduced when the nitriding temperature is higher than 300 °C. The formation of ε-Fe_3_N and γ-Fe_4_N will improve the hardness of stainless steel. As shown in Figure 7c, the 17-4PH sample more easily generated the CrN phase than did the 1Cr15 sample, which meant that 17-4PH stainless steel will be more prone to lose corrosion resistance.

#### 3.2.2. Microhardness Analysis

As shown in Figure 8, the surface hardness of the two stainless steels was gradually enhanced with increasing nitriding temperature or extended nitriding time. Under the same temperature and nitriding time, the surface hardness of the 1Cr15 sample was significantly higher than that of the 17-4PH sample. After nitriding at different temperatures for 24 h, the surface microhardness of 1Cr15 stainless steel was 2.05–3.48 times higher than that of the basic metal (600/HK_0.025_). After prolonging the nitriding time, the hardness of the sample surface was also improved, but the range of increase was low.

It is known from Fick’s law that the more intense the molecular movement and the faster the diffusion, the deeper the impact distance will be found on the higher temperature when the diffusion activation energy of the material is specific. Extending the nitriding time will make the thickness of the nitriding layer closer to the theoretical thickness at that temperature and simultaneously make the diffusion more adequate. The infiltrated N elements will exist in the lattice interstitial positions. The reaction has more time to precipitate at the grain boundaries combined with the metal elements to form a dispersive phase. The newly generated dispersion phase and the N atoms in the interstitial positions will produce dispersion and solid solution strengthening effects. It will impede dislocation movement, forming dislocation plugging and entanglement, enhancing the hardness and surface strength of the material. It can be seen from Figure 8b,c that the nitriding layer depth of the cross-sectional hardness was maintained increasingly deeper with nitriding temperature rises or extended nitriding time. However, hardness extended growth in the direction of the layer depth within 50 µm at the nitriding temperature of >350 °C. It may be that the surface layer is a porous compound layer that has a lower hardness than the nitrogen-diffusion zone beneath the surface layer. Therefore, the hardness of the sample showed first showed an increase and then a decrease. Simultaneously, the higher nitriding temperature, the longer the cooling process and the more uniform the diffusion process. It also formed a large diffusion equilibrium zone, which made the hardness stable in a larger depth range.

#### 3.2.3. Corrosion Resistance Behavior

Figure 9 shows the surface morphology of the 1Cr15 and 17-4PH stainless steel substrates with different nitriding conditions after a 360 h salt spray corrosion test. The BM and specimens of 1Cr15 treated at 250 °C for 24 h and 300 °C for 24 h were free of corrosion products on the surface and had good corrosion resistance (Figure 9a–c). However, salt spray corrosion resistance in nitriding specimens declined with increasing holding time and nitriding temperature. The nitriding specimens of 1Cr15 treated at 300 °C for 48 h produced local pitting after salt spray corrosion (Figure 10b), and nitriding specimens treated at 350–450 °C explained the large-scale comprehensive corrosion and yellow-brown corrosion products on the surface (Figure 9e–g). By comparing the 17-4PH and 1Cr15 samples under the same nitriding conditions, it can be seen that the corrosion resistance of 17-4PH after nitriding was lower than that of the 1Cr15 sample.

It can be seen from Figure 9a–c and Table 6 that with increasing nitriding temperature, the self-corrosion potential of the 1Cr15 sample became increasingly lower, the corrosion current density generally increased, and the corrosion resistance became worse. The main reason was that with increasing nitriding temperature, the CrN phase gradually precipitated from the surface nitriding layer, resulting in a gradual decline in corrosion resistance of the stainless steel. Simultaneously, after plasma nitriding at 300 °C for 24 h and 48 h, respectively, the corrosion current density of the samples decreased, while the change in the self-corrosion potential was not significant. Combining the XRD (Figure 7b,c) and SEM results (Figure 5b–e), it can be seen that obvious nitride particles appeared in the precipitates at the grain boundary of the nitriding sample at 350 °C for 24 h. Although the corrosion tendency was small, the influence of Cr deficiency on the material was dominant after corrosion, resulting in more serious corrosion. By comparing 1Cr15 and 17-4PH stainless steels under the same conditions, it can be seen that their salt spray corrosion experimental results were consistent. In other words, the corrosion resistance of 17-4PH was much lower than that of 1Cr15. DFT calculation shows that the corrosion resistance of stainless steel containing Mo is better than that of stainless steel containing Nb, which is also confirmed by electrochemical tests (Figure 11 and Table 6) and salt spray corrosion experiments. Simultaneously, the self-corrosion potential and self-corrosion current density of the two materials were highly consistent with the information reflected by the Fermi level calculated by DFT (Table 4 and Figure 4).The higher the Fermi energy level, the more easily the material is corroded, which also means that its self corrosion potential is lower and self corrosion current density is higher. It can be seen that the addition of molybdenum could also greatly improve the corrosion resistance of the material.

Figure 12 shows the nitriding samples’ Nyquist diagrams obtained at open circuit potential after soaking in 3.5 wt.% NaCl solution for 5 h. All Nyquist diagrams showed obvious capacitive reactance arcs, indicating that the corrosion process was controlled by charge transfer. The radius of the capacitive reactance arc is related to the dissolution behavior of stainless steel in solution. The larger the capacitive reactance arc radius, the better the corrosion resistance [37]. It was found that the capacitive reactance arc radius of 1Cr15 stainless steel was larger than that of the matrix after 24 h of low-temperature nitriding at 250 °C to 300 °C. It showed that under this condition, nitriding treatment improved the corrosion resistance of 1Cr15 stainless steel. For 17-4PH samples, the capacitive reactance arc was much lower than that of 1Cr15. Although the corrosion resistance of 17-4PH could be improved by prolonging the nitriding time at 300 °C, the corrosion resistance of the sample was still lower than that of 1Cr15 nitrided at the same temperature.

A Nyquist impedance spectrum comprises the capacitance loop caused by the charge transfer resistance and double-layer capacitance. The equivalent circuit of the two kinds of stainless steel obtained by model fitting is shown in Figure 13, where Rs is the solution resistance and Rct is the charge transfer resistance. Due to the differences in roughness of the electrode surface on the microscale and energy dissipation, the constant phase angle element CPE was used to simulate the electric double layer. The corrosion rate of the material was related to the charge transfer resistance Rct, which represented the trend of the sample surface providing electrons and releasing cations to the solution. The larger the Rct value, the harder the charge transfer, which meant that the more difficult the corrosion, the better the corrosion resistance of the material [38]. Table 7 shows the fitting results of the parameters of each element of the impedance equivalent circuit. By analyzing the change in Rct, it can be seen that nitriding of 1Cr15 stainless steel below 300 °C improved the material’s corrosion resistance, and the Rct value of 1Cr15 was greater than that of 17-4PH under the same conditions. Therefore, 1Cr15 had better corrosion resistance than 17-4PH.

## 4. Conclusions

(1)1Cr15Ni2Mo2Cu precipitation hardening steel containing Mo had better corrosion resistance than 17-4PH precipitation hardening stainless steel containing Nb. The former still had better corrosion resistance after low temperature nitriding treatment. This was attributed to the higher stability, lower Fermi level, and lower electron conduction concentration and electrochemical activity of stainless steel containing molybdenum than stainless steel containing niobium.(2)At the same ion nitriding temperature, the surface hardness of 1Cr15Ni2Mo2Cu steel increased linearly, and the corrosion resistance decreased when the nitriding time was prolonged. Under the same nitriding time (24 h), the nitriding temperature increased from 300 °C to 450 °C, and the surface hardness and nitriding layer depth of nitriding steel increased gradually, but the corrosion resistance decreased gradually. This was attributed to the Cr-poor phenomenon caused by the formation of CrN in the nitride layer.(3)The surface hardness of 1Cr15Ni2Mo2Cu steel could be significantly improved and good corrosion resistance could be maintained by ion nitriding treatment at 300 °C for 24 h.

## Figures and Tables

**Figure 1 materials-15-06575-f001:**
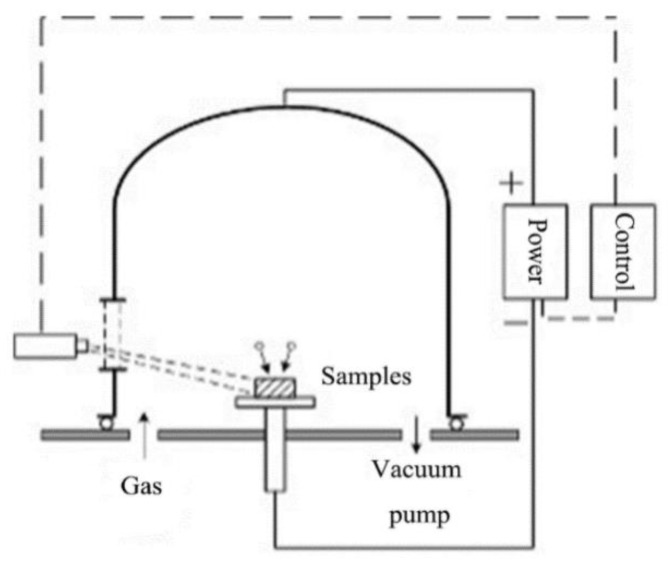
Schematic diagram of low-temperature plasma nitriding device.

**Figure 2 materials-15-06575-f002:**
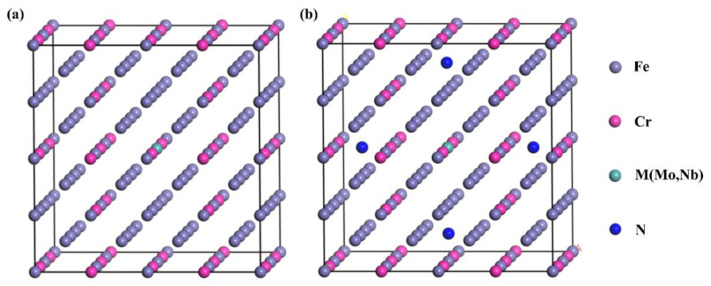
The crystal structure of (**a**) Fe105Cr22M and (**b**) Fe105Cr22MN4 (M = Mo or Nb) solid solution.

**Figure 3 materials-15-06575-f003:**
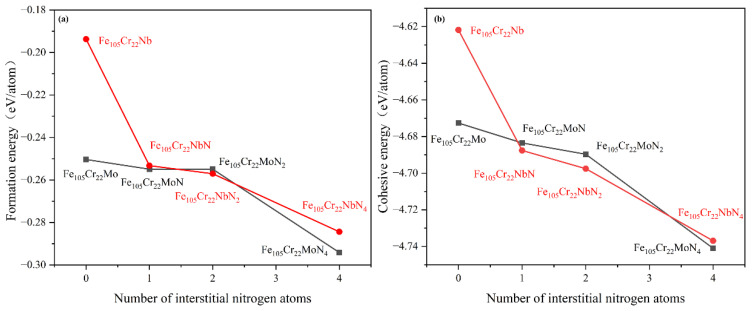
Variation of (**a**) formation energy and (**b**) cohesion energy of stainless steel with different nitriding degrees.

**Figure 4 materials-15-06575-f004:**
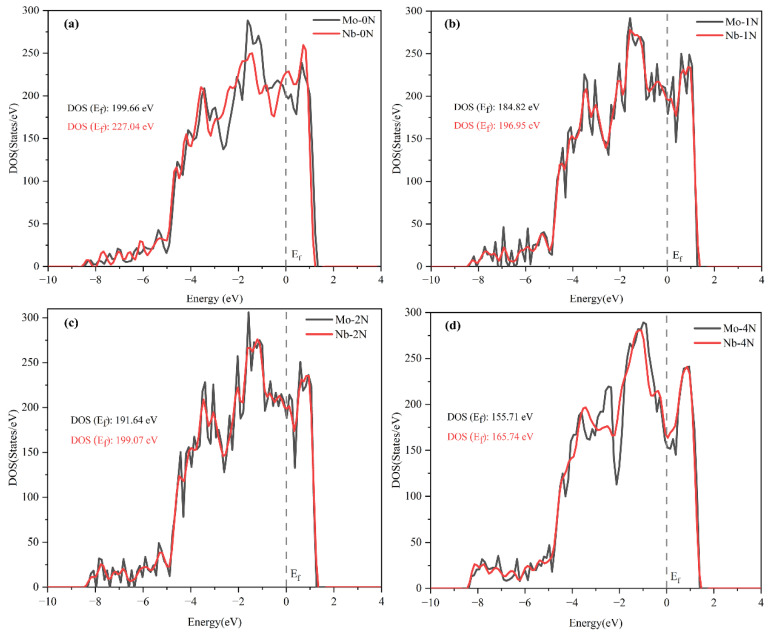
DOS of stainless steels with different nitriding degrees: (**a**) Fe105Cr22M; (**b**) Fe105Cr22MN1; (**c**) Fe105Cr22MN2; (**d**) Fe105Cr22MN4 (M = Mo or Nb).

**Figure 5 materials-15-06575-f005:**
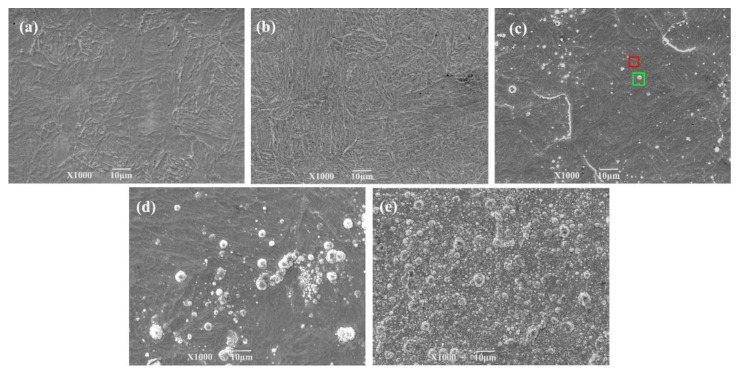
Surface morphology of 1Cr15 stainless steel after ion nitriding treatment: (**a**) 250 °C for 24 h; (**b**) 300 °C for 24 h; (**c**) 350 °C for 24 h (The rectangle is the EDS scanning point); (**d**) 400 °C for 24 h; (**e**) 450 °C for 24 h.

**Figure 6 materials-15-06575-f006:**
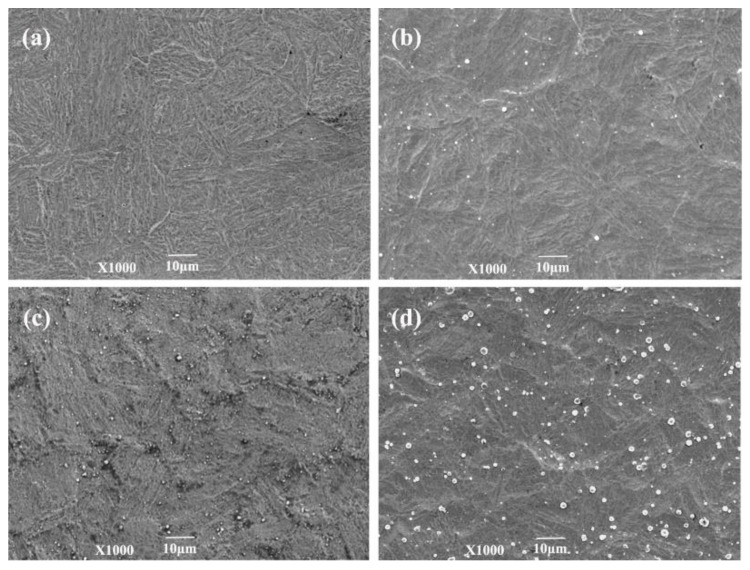
Surface morphology of two stainless steels after ion nitriding treatment: (**a**) 300 °C for 24 h (1Cr15); (**b**) 300 °C for 48 h (1Cr15); (**c**) 300 °C for 24 h (17-4PH); (**d**) 300° C for 48 h (17-4PH).

**Figure 7 materials-15-06575-f007:**
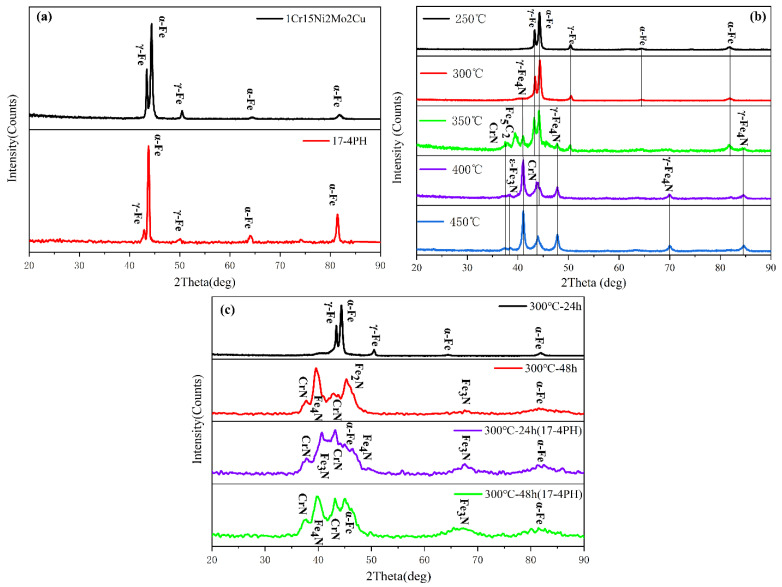
XRD curves of stainless steel by nitriding treatment of different conditions: (**a**) BM; (**b**) 1Cr15 nitriding for 24 h at different temperatures; (**c**) 17-4PH and 1Cr15 nitriding at 300 °C for different times.

**Figure 8 materials-15-06575-f008:**
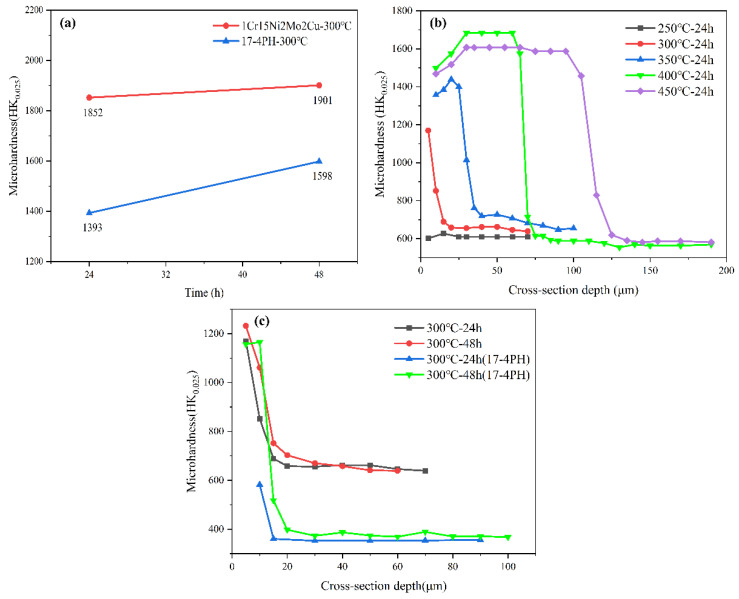
Surface and section hardness of stainless steel after plasma nitriding. (**a**) Surface hardness of nitrided samples at different temperatures and holding times. (**b**) Section hardness of 1Cr15 nitrided samples at different temperatures. (**c**) Section hardness of nitrided samples at different temperatures and holding times.

**Figure 9 materials-15-06575-f009:**
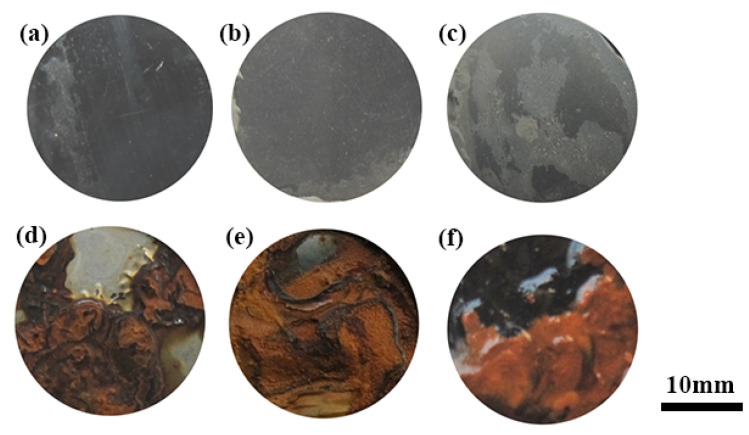
Morphology of 1Cr15 stainless steels after salt spray corrosion of ion nitriding samples for 360 h under different conditions: (**a**) BM; (**b**) 250 °C for 24 h; (**c**) 300 °C for 24 h; (**d**) 350 °C for 24 h; (**e**) 400 °C for 24 h; (**f**) 450 °C for 24 h.

**Figure 10 materials-15-06575-f010:**
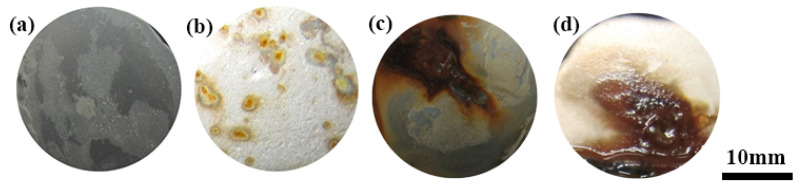
Morphology of two kinds of stainless steels after salt spray corrosion of ion nitriding samples for 360 h under different conditions: (**a**) 300 °C for 24 h (1Cr15); (**b**) 300 °C for 48 h (1Cr15); (**c**) 300 °C for 24 h (17-4PH); (**d**) 300 °C for 48 h (17-4PH).

**Figure 11 materials-15-06575-f011:**
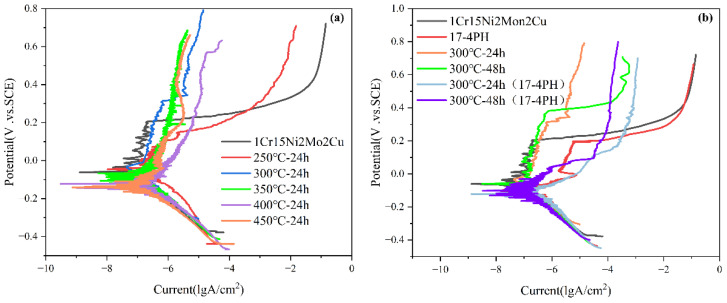
Electrochemical polarization curves of (**a**) 1Cr15 stainless steel at different nitriding temperatures and (**b**) 1Cr15 and 17-4PH stainless steel at different nitriding times in 3.5% NaCl solution.

**Figure 12 materials-15-06575-f012:**
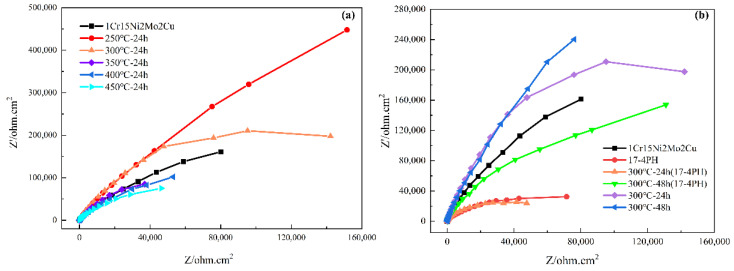
Nyquist impedance spectra of (**a**) 1Cr15 stainless steel at different nitriding temperatures and (**b**) 1Cr15 and 17-4PH stainless steel at different nitriding times.

**Figure 13 materials-15-06575-f013:**
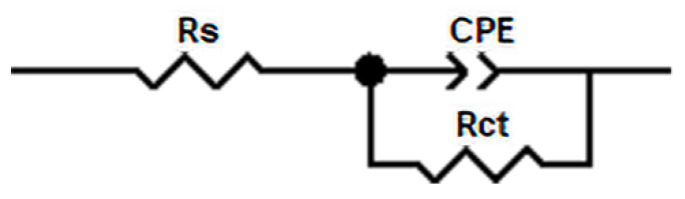
Impedance equivalent circuit diagram of stainless steel with different nitriding times.

**Table 1 materials-15-06575-t001:** Chemical composition (wt.%).

Steels	C	Si	Mn	Mo	Nb	Ni	Cr	Cu	S	P	Fe
1Cr15	0.14	1.0	1.0	2	—	2	16.5	0.75	0.03	0.02	Bal.
17-4PH	0.07	0.4	0.74	—	0.5	4.23	16.92	3.44	0.03	0.03	Bal.

**Table 2 materials-15-06575-t002:** Mechanical properties of stainless steel.

Material	$\sigma_{s}/\mathrm{MPa}$	$\sigma_{b}/\mathrm{MPa}$	$\delta_{5}/\%$	ψ/%	Hardness/HRC
1Cr15	1030	1500	15.3	50.2	55
17-4PH	970	1012	17.4	66.3	37

**Table 3 materials-15-06575-t003:** Total, cohesive and formation energies of systems with different substitutional solid solutions.

System	*E_tot_*/eV	E_coh_/eV	E_form_/eV
Fe105Cr22Mo	−1067.451	−4.672	−0.250
Fe105Cr22MoN	−1076.635	−4.683	−0.254
Fe105Cr22MoN2	−1084.695	−4.689	−0.254
Fe105Cr22MoN4	−1107.624	−4.741	−0.294
Fe105Cr22Nb	−1059.471	−4.621	−0.193
Fe105Cr22NbN	−1075.693	−4.687	−0.253
Fe105Cr22NbN2	−1084.790	−4.697	−0.257
Fe105Cr22NbN4	−1105.607	−4.736	−0.284

**Table 4 materials-15-06575-t004:** Fermi energy levels of stainless steel.

System	E_f_/eV
Fe105Cr22Mo	7.599
Fe105Cr22MoN	7.628
Fe105Cr22MoN2	7.633
Fe105Cr22MoN4	7.644
Fe105Cr22Nb	7.632
Fe105Cr22NbN	7.626
Fe105Cr22NbN2	7.635
Fe105Cr22NbN4	7.668

**Table 5 materials-15-06575-t005:** EDS data of nitrided layer surface (wt.%).

Sign	N	Cr	Fe	Ni	Mo
Precipitate area (green rectangle)	9.67	16.02	72.91	3.63	1.60
Smooth area (red rectangle)	6.22	16.13	69.02	3.37	1.37

**Table 6 materials-15-06575-t006:** Relevant electrochemical parameters calculated by Tafel fitting.

Nitriding Temperature and Times	E_corr_ (V vs. SCE)	I_corr_ (μA/ cm^2^)
1Cr15	−0.061	0.0463
17-4PH	−0.104	0.218
250 °C–24 h	−0.0468	0.128
300 °C–24 h	−0.0409	0.103
350 °C–24 h	−0.0974	0.188
400 °C–24 h	−0.135	0.233
450 °C–24 h	−0.139	0.204
300 °C–24 h	−0.0409	0.103
300 °C–48 h	−0.060	0.0389
300 °C–24 h (17-4PH)	−0.111	0.267
300 °C–48 h (17-4PH)	−0.099	0.0617

**Table 7 materials-15-06575-t007:** Fitting parameters of components of EIS equivalent circuit.

Steels	Rs/(Ω·cm^2^)	CPE		Rct/(Ω·cm^2^)
		Y0/(S·sec^n^·cm^−2^)	n	
1Cr15	4.508	2.569 × 10^−5^	0.9093	7.127 × 10^5^
17-4PH	5.088	5.324 × 10^−5^	0.8749	4.551 × 10^5^
250 °C–24 h	11.30	2.441 × 10^−5^	0.9133	2.156 × 10^6^
300 °C–24 h	4.506	3.004 × 10^−5^	0.9145	7.500 × 10^5^
350 °C–24 h	8.847	1.074 × 10^−4^	0.8593	1.546 × 10^5^
400 °C–24 h	6.111	8.972 × 10^−5^	0.8658	3.180 × 10^5^
450 °C–24 h	33.63	1.298 × 10^−4^	0.8663	1.038 × 10^5^
300 °C–24 h	4.506	3.004 × 10^−5^	0.9145	7.500 × 10^5^
300 °C–48 h	4.194	4.188 × 10^−5^	0.9135	1.130 × 10^6^
300 °C–24 h (17-4PH)	4.162	5.061 × 10^−5^	0.8904	1.074 × 10^5^
300 °C–48 h (17-4PH)	4.579	4.156 × 10^−5^	0.8917	5.242 × 10^5^

## Data Availability

Not applicable.

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
