# Peer review of "Effect of Alloying Elements and Low Temperature Plasma Nitriding on Corrosion Resistance of Stainless Steel"

_materials, 2022, doi:10.3390/ma15196575_

Round 1
Reviewer 1 Report
In this paper, the authors evaluated the effects of Mo and Nb on the corrosion resistance of martensitic stainless steel. Overall, the paper is well organized, and the conclusions are sound. The major problem is the missing experimental details and some figures' bad quality, which should be addressed before publication. The detailed comments are as follows:
1. How were the data in Table 4 collected?
2. Figure 4: It is better to add the exact DOS value at the Fermi level on both curves.
3. Figure 5(g): the words in the figure are difficult to read.
4. Line 285: How confident are the authors about the existence of ε-Cu? I suspect the "ε-Cu" diffraction peaks are actually austenite peaks, considering the low Cu concentration in 1Cr15 alloy. Besides, I have never seen anyone report the Cu peaks in 17-4ph XRD patterns. Please double-check this with caution.
5. Figure 7: The (a), (b), (c), and (d) panels were not indicated in the figure. Most importantly, this paper did not provide any experimental details and parameters for the XRD experiments.
6. Figure 7(a): The 17-4ph XRD intensity is too low to be read.
7. Lines 306-307: The resolution of Figure 7(c) is not good enough to support this argument. The clear peaks of CrN cannot be seen from the figure.
8. Section 3.3.2 is not relevant to the topic of this paper and thus should be removed.
9. Figure 10(a): How many data points were collected on each sample? What is the standard deviation of the measurements?
10. Lines 364-367: The discussion may not be correct. Most likely, the reason that the surface layer has a lower hardness than the inner layer is that the surface layer is a porous White Layer (compound layer), which has a lower hardness than the Nitrogen-diffusion zone beneath the surface layer.
11. Figure 12: The data is too "noisy" for display.
Other comments:
1. Lines 103-104: grammar check.
2. Line 134: Is it “mHz” or “MHz”?
3. Line 143: It is not necessary to cite so many references.
4. Line 152: The martensite stainless steel has α-Fe (bcc) as the main lattice structure, not the phase.
Reviewer 2 Report
the manuscript is of interests to read. however, there are a couple of issues needed to be addressed.
1) modify figure 5g: it is hard to read the onset information.
2) figure 12 and table 5 should plot potential with respect to reference electrode type.
Reviewer 3 Report
There are the following comments on the material of the article:
1. The authors compare two very different chemical composition of steel grades. In these steels, there are different amounts of not only common constituent components (Ni, Cu, C, Si), but also alloying (Mo, Nb). At the same time, the authors write (lines 93 and 94): "The difference between these two kinds of stainless steels was whether they contained molybdenum or niobium." But the difference in the compositions is significant. It would be more logical to conduct a study with a number of steels of the same qualitative, but different quantitative compositions, or with steels with the same amounts of main components (Ni, Cu, C, Si) and different amounts of alloying (Mo, Nb). At least within the designated title of the article. The selected steels also have serious differences in physical and mechanical characteristics, so it is necessary to give a detailed justification for the choice of steel grades, which at the moment is not clear.
2. In Figures 5 and 6, it is necessary to leave only the images obtained with the same low temperature plasma nitriding treatment modes. Or add pictures to Figure 6 for other steel surface treatment modes. A similar remark applies to Figures 8 and 9, as well as to Figure 11 and tables 5 and 6. There is no correlation of data in the current format of the presentation of the material, since the data is mixed, they need to be separated and systematized.
3. In Figure 10 (c), leave the data only for steel 1 Cr 15. In a separate figure (10 - d), take out the data for steels under the same conditions (300 °C - 24 and 48 h) and make a comparison, as is done in Figure 12. A similar note to Figure 13 (b).
4. Lines 412-414: "Simultaneously, the self-corrosion potential and self-corrosion current density of the two materials were highly consistent with the information reflected by the Fermi level calculated by DFT."
What is this "highly consistent"? In what data is the relationship and correspondence of the data traced? It is necessary to show this in the text, since it remains unclear why paragraph 3.1 is given.
The article requires significant revision.
Round 2
Reviewer 1 Report
The authors did a good job addressing my comments. I am satisfied with the revision. Congratulations!
Reviewer 3 Report
Corrections accepted.